# A NOTCH3 pathogenic variant influences osteogenesis and can be targeted by antisense oligonucleotides in induced pluripotent stem cells

Ernesto Canalis[1,2,3]*, Jungeun Yu[1,3¤], Lauren Schilling[3], Paymaan Jafar-nejad[4], Michele Carrer[4]

1 Departments of Orthopaedic Surgery, UConn Musculoskeletal Institute, UConn Health, Farmington, CT, United States of America, 2 Departments of Medicine, UConn Musculoskeletal Institute, UConn Health, Farmington, CT, United States of America, 3 UConn Musculoskeletal Institute, UConn Health, Farmington, CT, United States of America, 4 Ionis Pharmaceuticals, Inc., Carlsbad, CA, United States of America

¤ Current address: Cellinfinity Bio, West Haven, CT, United States of America
* canalis@uchc.edu

**Data Availability Statement:** All relevant data are within the manuscript.

**Funding:** This work was supported by the National Institute of Arthritis and Musculoskeletal and Skin

## Abstract

Lateral Meningocele Syndrome (LMS), a disorder associated with *NOTCH3* pathogenic variants, presents with neurological, craniofacial and skeletal abnormalities. Mouse models of the disease exhibit osteopenia that is ameliorated by the administration of Notch3 antisense oligonucleotides (ASO) targeting either *Notch3* or the *Notch3* mutation. To determine the consequences of LMS pathogenic variants in human cells and whether they can be targeted by ASOs, induced pluripotent NCRM1 and NCRM5 stem (iPS) cells harboring a *NOTCH3^{6692-93insC}* insertion were created. Parental iPSCs, *NOTCH3^{6692-93insC}* and iso-genic controls, free of chromosomal aberrations as determined by human CytoSNP850 array, were cultured under conditions of neural crest, mesenchymal and osteogenic cell differentiation. The expected cell phenotype was confirmed by surface markers and a decline in *OCT3/4* and *NANOG* mRNA. *NOTCH3^{6692-93insC}* cells displayed enhanced expression of Notch target genes *HES1*, *HEY1*, *2* and *L* demonstrating a NOTCH3 gain-of-function. There was enhanced osteogenesis in *NOTCH3^{6692-93insC}* cells as evidenced by increased mineralized nodule formation and *ALPL*, *BGLAP* and *BSP* expression. ASOs targeting *NOTCH3* decreased both *NOTCH3* wild type and *NOTCH3^{6692-93insC}* mutant mRNA by 40% in mesenchymal and 90% in osteogenic cells. ASOs targeting the *NOTCH3* insertion decreased *NOTCH3^{6692-93insC}* by 70–80% in mesenchymal cells and by 45–55% in osteogenic cells and *NOTCH3* mRNA by 15–30% and 20–40%, respectively. In conclusion, a *NOTCH3* pathogenic variant causes a modest increase in osteoblastogenesis in human iPS cells *in vitro* and NOTCH3 and NOTCH3 mutant specific ASOs downregulate *NOTCH3* transcripts associated with LMS.

Diseases (NIAMS) [AR076747 (EC) and AR072987 (EC)]. The content is solely the responsibility of the authors and does not necessarily represent the official views of the National Institutes of Health. he funders had no role in study design, data collection and analysis, decision to publish, or preparation of the manuscript.

**Competing interests:** PJ and MC are paid employees of Ionis Pharmaceuticals. Please note that the synthesis and applications of antisense oligonucleotides may be covered by patent(s) filed by Ionis Pharmaceuticals. Individuals wanting to obtain antisense oligonucleotides from Ionis Pharmaceuticals are required to contact Ionis Pharmaceuticals directly. This does not alter our adherence to PLOS ONE policies on sharing data and materials." as detailed in http://journals.plos.org/plosone/s/competing-interests

**Abbreviations:** The abbreviations used are ASO, antisense oligonucleotides; bp, base pair; cEt, constrained ethyl; DMEM, Dulbecco's modified Eagle's medium; FACS, fluorescence activated cell sorting; GSKi, glycogen synthase inhibitor; iPS, induced pluripotent stem; LMS, Lateral Meningocele Syndrome; MSC, mesenchymal cells; MOE, methoxyethyl; NCRM, NIH control reference line; NRR, negative regulatory region; NC, neural crest; NICD, NOTCH3 intracellular domain; PCR, polymerase chain reaction; PEST, proline (P)-, glutamic acid (E)-, serine (S)-, threonine (T)-rich; qRT-PCR, quantitative reverse transcription-polymerase chain reaction; RANKL, receptor activator of NF-κB ligand; sg, single guide; TGF, transforming growth factor.

## Introduction

Lehman Syndrome or Lateral Meningocele Syndrome (LMS) (OMIM 130720) is a rare and devastating disorder characterized by meningoceles and numerous skeletal manifestations, including craniofacial developmental defects, short stature, scoliosis and osteopenia [1–3]. The syndrome is associated with mutations, insertions or short deletions in exon 33 of *NOTCH3* upstream of the proline (P)-, glutamic acid (E)-, serine (S)-, threonine (T)-rich (PEST) domain [4, 5]. The mutations lead to the premature termination of a protein product lacking sequences required for the ubiquitination and degradation of the NOTCH3 intracellular domain (NICD) so that the protein is stable and a gain-of-NOTCH3 function ensues. The inheritance of the disorder is autosomal dominant although de novo heterozygous mutations may occur [2]. The incidence of LMS is unknown and less than 100 cases have been reported. Treatment for this genetic disorder affecting the skeleton is not available.

LMS is one of many disorders that are associated with specific gene mutations that present with devastating clinical manifestations [6, 7]. Unfortunately, there is no practical or effective intervention that corrects the genetic abnormality, leading to the unsuccessful management of subjects afflicted by these disorders. Gene editing has been proposed to correct mutations in mice and humans [8, 9]. However, gene editing is not readily available for therapeutic intervention, and ethical concerns have been raised regarding genome editing in human embryos [10–12]. A specific tissue could be targeted with vectors with preferential affinity for the tissue affected to replace the mutant DNA with repaired DNA, an approach that also has been challenging [13].

Antisense oligonucleotides (ASO) are a novel and potential therapeutic approach to downregulate wild type and mutant transcripts. ASOs have been successful in the targeting and downregulation of mutant genes in the liver, central and peripheral nervous system and retina [14–20]. ASOs are synthetic single-stranded nucleic acids that by binding to target mRNA by Watson-Crick pairing they cause the degradation of mRNA by RNase H [21, 22]. As such, ASOs are of potential value in conditions where a gain-of-function exists since gene downregulation could correct the functional outcome. Previously, we demonstrated the effectiveness of ASOs targeting either wild type or mutant *Notch3* in mice and skeletal cells from mice harboring a *Notch3* mutation resulting in a gain-of-function analogous to the one observed in LMS [23, 24].

An attractive application of induced pluripotent stem (iPS) cell technology is that it allows the isolation of patient-derived cells that carry the genetic alterations associated with specific disorders or the creation of stocks of established iPS cell lines harboring specific mutations. iPS cells provide an experimental system not only to study the pathogenesis of the disease but also to devise therapeutic strategies [25]. Although the mutations present in individuals with LMS are in exon 33 of *NOTCH3*, not every family line harbors the same mutation, and this is often the case with monogenic disorders [4]. Therefore, ASOs targeting each mutation would need to be designed and tested for their effectiveness in downregulating the *NOTCH3* mutant transcript. This would require ample supply of cells from afflicted individuals so that ASOs specific to the pathogenic variant can be tested to document efficacy. A practical approach is the establishment of stocks of iPS cells from affected individuals or the introduction of specific mutations in stocks of iPS cells to be tested for ASO effectiveness on mutant transcript downregulation and cellular behavior.

The intent of the current study was to create and establish iPS cell lines harboring a *NOTCH3* pathogenic variant found in LMS and define their cellular behavior and differentiation to the osteogenic lineage. In addition, ASOs targeting either the wild type or *NOTCH3* pathogenic variant were tested to determine their effectiveness in downregulating *NOTCH3*.

## Materials and methods

### Induced pluripotent stem (iPS) cells

NIH control reference line (NCRM)1 and NCRM5 were generated by the NIH Regenerative Medicine Program and obtained through RUCDR Infinite Biologics (Piscataway, NJ) [26]. NCRM1 and NCRM5 cells are derived from male CD14+ cord blood by episomal integration and are well-characterized pluripotent cells devoid of chromosomal structural aberrations.

Human *NOTCH3* mutant iPS cells were created at the University of Connecticut Cell and Engineering Core (Farmington, CT). To introduce the *NOTCH3*$^{6692-93insC}$ insertion into *NOTCH3*, the databases http://www.rgenome.net and https://chopchop.cbu.uib.no were utilized to evaluate potential single guide (sg)RNAs. NOTCH3 sgRNA 5'-AGAGGTCAAGGCC AGGACTA-3' was selected because of its high score with minimum off-target effects. This guide RNA targets the intron upstream of exon 33 containing the mutation and was cloned into pSpCas9(BB)-2A-Puro (PX459v2, Addgene 62988, Watertown, MA). Homology arms flanking a neomycin selection cassette flanked by loxP sites were cloned into a targeting vector so that the guide RNA sequence would be split by the cassette to prevent further DNA cleavage following the targeting of the 3' homology arm contained either in the *NOTCH3*$^{6692-93insC}$ mutant or the wild type sequence to be targeted (Fig 1). NCRM1 and NCRM5 iPS cells were nucleofected using a Lonza 4d Nucleofector and primary kit P3 following manufacturer's protocols (Lonza, Basel, Switzerland). Cells were grown and clones established, and genomic DNA was screened for vector-targeting and positive clones were analyzed by DNA sequencing to determine genotype and the presence or absence of the *NOTCH3*$^{6692-93insC}$ insertion. Depending on the breakpoint occurring during homologous directed repair, cells could harbor the mutation, if the breakpoint was at the end of the homology arm or harbor a wild type allele

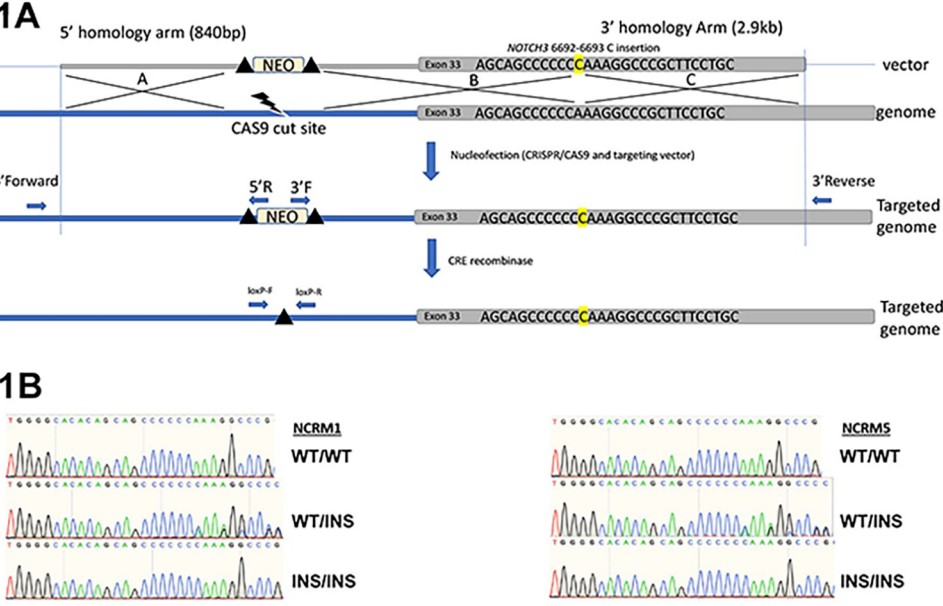

**Fig 1. Targeting of *NOTCH3* and creation of *NOTCH3*$^{6692\_93insC}$ mutant induced pluripotent stem (iPS) cells.** In Panel A, NCRM1 and NCRM5 cells were nucleofected with pSPCas9(BB)-2A-Puro to deliver Cas9 and the targeting vector, containing the *NOTCH3*$^{6692-93insC}$ insertion depicted above in Band C or isogenic control. The neo selection cassette (in A) was deleted following nucleofection of Cre-IRES-PuroR. Breakpoints in homologous directed repair in the homology arm (C) would include the mutation and breakpoint in the insertion (B) would create a wild type allele. In Panel B, DNA sequencing of *Notch3*$^{6692-93insC}$ homozygous and heterozygous mutant and *NOTCH3* isogenic control NCRM1 and NCRM5 iPS cells.

if the breakpoint occurred at the insertion site. Clones were transfected with a Cre-IRE-S-PuroR vector (Addgene 30205) to delete the neomycin selection cassette and loxP recombination verified by polymerase chain reaction (PCR). Homozygous, and heterozygous clones and wild type isogenic controls were obtained. Absence of genetic aberrations was verified using the Infinium CytoSNP-850K v1.3 BeadChip microarray (Illumina; San Diego, CA) at the University of Connecticut Center for Genome Innovation (Storrs, CT) in accordance with manufacturer's instructions. Parental NCRM1 and *NOTCH3* mutant and isogenic control NCRM1 cells harbored a 602.9 kilobase gain in chromosome 20 Band 20q11.21-20q11.21 position 29, 804, 293–30, 407, 232.

## iPS cell culture and *in vitro* osteogenesis

NCRM1 and NCRM5 iPS cells with and without the *NOTCH3*$^{6692-93insC}$ insertion were seeded on matrigel-coated plates (STEMCELL Technologies, Vancouver, Canada or Geltrex, Waltham, MA) and cultured in mTeSR Plus (STEMCELL Technologies) medium at 37˚C in a 5% $CO_2$ atmosphere and manually passaged twice a week with the aid of a 28-gauge needle using an enzyme-free cell dissociation solution at 37˚C and streaked through the well for colony formation [26–28]. For neural crest differentiation, cells were cultured on laminin-521 (Thermo Fisher, Waltham, MA) for 6 days and the culture medium was replaced twice a week with basal STEMdiff™ APEL™ differentiation medium (STEMCELL Technologies) supplemented with 5 µM glycogen synthase inhibitor (GSKi) (CHIR992021, Stemgent, Lexington, MA) and 10 µM transforming growth factor (TGF)β inhibitor (SB431542, Sigma-Aldrich, St. Louis, MO) (Fig 2) [27, 28]. Cells were detached by adding Stempro Accutase (Thermo Fisher) and selected for CD34⁻KDR⁻CD271⁺PDGFRa⁻ surface markers by fluorescence activated cell sorting (FACS) [29]. Following sorting, cells were plated at a 1–2 x 10⁶ cells/10 cm² density on laminin-521 (Thermo Fisher) coated dishes in the presence of STEMdiff™ APEL™ Medium, 10 µM TGFβ inhibitor, 10 µM Rho kinase (ROCK) inhibitor (Y-27632, STEMCELL Technologies) and antibiotic-antimycotic cocktail (Thermo Fisher) for 24 h. To induce mesenchymal cell differentiation, cells were cultured for one week on laminin-521 coated plates and for one week on Cell Bind coated plates (Thermo Fisher) in MSC NutriStem XF medium (Sartorius, Gottingen, Germany) [28]. Mesenchymal cell differentiation was verified by the presence of CD44⁺CD73⁺CD105⁺CD31⁻CD45⁻ surface markers by flow cytometry.

To induce osteogenic differentiation, mesenchymal cells were grown to confluence on Cell Bind coated plates in NutriStem XF medium and switched to Dulbecco's modified Eagle's medium (DMEM) (Thermo Fisher), 10% fetal bovine serum (Atlanta Biologicals, Laurenceville, GA), 100 nM dexamethasone, 50 ng/ml ascorbic acid, 10 mM β-glycerophosphate (all from Sigma-Aldrich) [30]. Medium was replaced twice a week and cells were cultured for up to 4 weeks. To determine the presence of mineralized nodules, cultured cells were fixed and stained with alizarin red at various times during the culture period.

## NOTCH3 antisense oligonucleotides (ASO)

ASOs targeting the *NOTCH3* or the *NOTCH3*$^{6692-93insC}$ mutant pre-mRNA and a non-targeting control ASO that does not hybridize to any specific mRNA in the human or mouse transcriptome sequence were designed and synthesized by Ionis Pharmaceuticals (Carlsbad, CA). The ASO targeting human *NOTCH3* has a central segment of eight DNA nucleotides flanked on each side by five nucleotides with 2′-O-(2-methoxyethyl) (MOE) ribose sugar modifications, and contains phosphodiester or phosphorothioate backbone linkages [31]. The ASOs targeting *NOTCH3*$^{6692-93insC}$ have a central segment of ten DNA nucleotides flanked on each side by three nucleotides with 2′4′-constrained 2′-O-ethyl (cEt) modifications and contain

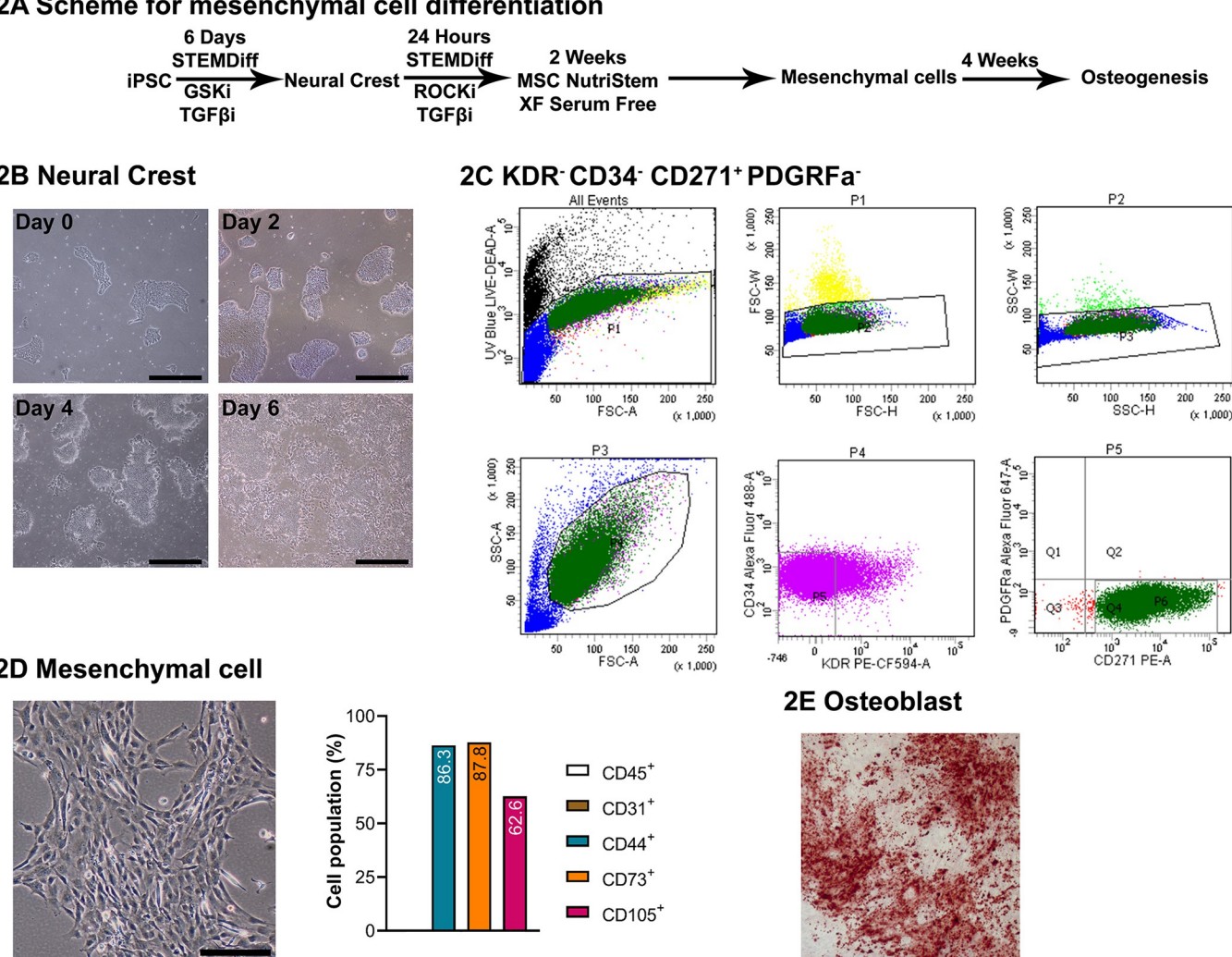

**Fig 2. NCRM1 and NCRM5 induced pluripotent stem (iPS) cells differentiate into neural crest and mesenchymal cell lineage *in vitro*.** iPS cells were cultured in STEMdiff™ APEL™ medium in the presence of GSK and TGFβ inhibitors, selected for KDR⁻CD34⁻CD271⁺PDGFRa⁻ by FACS and cultured in the presence of ROCK and TGFβ inhibitors for 24 h. Cells were transferred to Laminin-521 and Cell Bind coated plates in the presence of NutriStem XF medium for 2 weeks and osteogenic culture conditions for 4 weeks. Panel A shows the cell differentiation scheme. Panels B and C show the neural crest cells and surface markers following FACS. Panel D shows mesenchymal cells and surface markers determined by flow cytometry and Panel E shows mineralized nodules stained with alizarin red following osteogenesis. Bars in the right corner of images in B and E represent 500 μm, and in D 200 μm.

phosphorothioate backbone linkages. *NOTCH3* mutant or control ASOs at various doses and periods of time were added directly to mesenchymal cells before and after they were exposed to culture medium under osteogenic conditions for 2 weeks as indicated in text and legends.

## Quantitative reverse transcription-polymerase chain reaction (qRT-PCR)

Cellular RNA was extracted with the RNeasy kit (Qiagen, Valencia, CA), following manufacturer's protocols [32–35]. The iScript RT-PCR kit (BioRad, Hercules, CA) was used to reverse transcribe equal amounts of RNA which was then amplified in the presence of specific primers (IDT) (Table 1) with the SsoAdvanced Universal SYBR Green Supermix or the iQ SYBR Green Supermix (BioRad) at 60°C for 35 cycles. Transcript copy number was estimated by comparison with serial dilutions of cDNA of the genes examined obtained from Thermo

**Table 1. Primers used for qRT-PCR determinations.** GenBank accession numbers identify transcript recognized by primer pairs.

| Gene | Strand | Sequence | GenBank Accession Number |
|---|---|---|---|
| ACP5 | Forward | 5'–ATGAGAATGGCGTGGGCTAC–3' | NM_001111034.3 |
| | Reverse | 5'–GTGCCGCTTTGAGGGGTC–3' | |
| ALPL | Forward | 5'–GCAGACATTCTCAAAGCCTCTT–3' | NM_000478 |
| | Reverse | 5'–TCTGGAGAAATACGTTCGCTAGA–3' | |
| BGLAP | Forward | 5'–AAAGGTGCAGCCTTTGTGTC–3' | NM_199173 |
| | Reverse | 5'–GGTCTCTTCACTACCTCGCT–3' | |
| BSP | Forward | 5'–CGAGCCTATGAAGATGAGT–3' | NM_004967 |
| | Reverse | 5'–GGTGGTGGTAGTAATTCTGA–3' | |
| FOXC2 | Forward | 5'–TTGAGAACTCGACCCTCG–3' | NM_005251 |
| | Reverse | 5'–CTGCGTGGCGATAGAGAG–3' | |
| GSC | Forward | 5'–CAGACAGACGATGCTACT–3' | NM_173849 |
| | Reverse | 5'–TCCTCGTTCCTCTTTCTC–3' | |
| HES1 | Forward | 5'–CAACCCACCTCTCTTCCCTC–3' | NM_005524 |
| | Reverse | 5'–TTCTCTCCCAGTATTCAAGTTCCT–3' | |
| HEY1 | Forward | 5'–CCCAACTACATCTTCCCAGA–3' | NM_001040708 |
| | Reverse | 5'–TCTCAATTATTCCTCTCCGTCTT–3' | |
| HEY2 | Forward | 5'–GGGGTAAAGGCTACTTTGAC–3' | NM_012259 |
| | Reverse | 5'–AACTTCTGTTAGGCACTCTC–3' | |
| HEYL | Forward | 5'–GATCACTTGAAAATGCTCCAT–3' | NM_014571 |
| | Reverse | 5'–GGCATCAAAGAATCCTGTCC–3' | |
| HPRT | Forward | 5'–CCTGGCGTCGTGATTAGTG–3' | NM_000194.3 |
| | Reverse | 5'–CCCTTTCCAAATCCTCAGCATAA–3' | |
| NANOG | Forward | 5'–CAGTCTGGACACTGGCTGAA –3' | NLM_024865 |
| | Reverse | 5'–CTCGCTGATTAGGCTCCAAC –3' | |
| NOTCH1 | Forward | 5'–TGGACGACAACCAGAATGA–3' | NM_017617 |
| | Reverse | 5'–CCTCGAACCGGAACTTCTT–3' | |
| NOTCH2 | Forward | 5'–GGCTATGAACCCTGTGTAAA–3' | NM_024408 |
| | Reverse | 5'–CTTCTGGACATTTGCAGTATC–3' | |
| NOTCH3 | Forward | 5'–GAGGTCGTTGCACCCAGC–3' | NM_000435 |
| | Reverse | 5'–AGTGACAGGGGTCCTCCAG–3' | |
| NOTCH4 | Forward | 5'–GCATTGGTCTCAAGGCACTGAA–3' | NM_004557 |
| | Reverse | 5'–CCTGAGCACATCACAACTCCATC–3' | |
| OCT3/4 | Forward | 5'–TGTACTCCTCGGTCCCTTTC–3' | NM_002701 |
| | Reverse | 5'–TCCAGGTTTTCTTTCCCTAGC–3' | |
| RUNX2 | Forward | 5'–CAAGCAGAATTTAGCAGAGAT–3' | NM_001024630.4 |
| | Reverse | 5'–AGAAGGACCAGAGAACAAG–3' | |
| SOX10 | Forward | 5'–CCTCTCCTAGCCACTCTA–3' | NM_006941 |
| | Reverse | 5'–TTATTATGTGGAATGCTTAATGC–3' | |
| SP7 | Forward | 5'–CTCAACAACTCTGGGCAAAG–3' | NM_001173467 |
| | Reverse | 5'–GGAGGCTGAAAGGTCACT–3' | |
| WNT1 | Forward | 5'–CTTCGGCAAGATCGTCAA–3' | NM_005430.4 |
| | Reverse | 5'–GATGGAACCTTCTGAGCA–3' | |

Fisher Scientific or Dharmacon (Lafayette, CO). In experiments designed with the intent to determine an effect on *NOTCH3*[6692-93insC] transcripts by an ASO, fluorescent tagged products were used to conduct RT-PCR. Moloney murine leukemia virus reverse transcriptase was used to reverse transcribe total RNA and *NOTCH3* cDNA was amplified by qPCR in the presence

of SsoAdvanced Universal Probes Supermix (BioRad), *NOTCH3* and *NOTCH3* mutant primers and HEX labeled *NOTCH3* and FAM labeled *NOTCH3^(6692-93insC)* probes (BioRad) for 10 secs at 95°C followed by 30 secs at 60°C and repeated for 45 cycles [36]. Copy number of *NOTCH3* or *NOTCH3^(6692-93insC)* was estimated by comparison to a serial dilution of a 100 to 200 base pair (bp) synthetic DNA fragment (IDT) harboring (or not) the 6692-93insC in the *NOTCH3* locus cloned into pcDNA3.1(-) (Thermo Fisher) by isothermal single reaction assembly using reagents available commercially (New England BioLabs, Ipswich, MA) [37]. Transcripts are expressed as copy number corrected for *HPRT* expression.

## Statistics

Data are expressed as means ± SD and as individual sample values of technical replicates. Statistical significant differences were determined by either unpaired *t* test for pairwise comparisons or by two-way analysis of variance for multiple comparisons with Holm Šidák post-hoc analysis employing the GraphPad Prism version 10.2.0 for Windows 10 (GraphPad Software, San Diego, CA).

## Results

### Creation of *NOTCH3^(6692-93insC)* mutant cells

CRISPR/Cas9 technology was used to introduce a single base pair insertion at c.6692_93insC in *NOTCH3* in both NCRM1 and NCRM5 cells. The insertion predicts a premature termination upstream of the PEST domain of NOTCH3 duplicating the pathogenic variant found in a subject afflicted by LMS and functionally replicated in the *Notch3^(em1Ecan)* mouse mutant [38, 39]. NCRM1 and NCRM5 cells homozygous and heterozygous for the c.6692_93insC were obtained and verified by DNA sequencing and stocks created to be used for testing (Fig 1). Isogenic controls and heterozygous and homozygous *NOTCH3^(6692-93insC)* cells retained genomic stability and were absent of chromosomal aberrations as determined by the human CytoSNP-850 array.

### NCRM1 and NCRM5 cells differentiate into mesenchymal and osteogenic cells *in vitro*

Parental NCRM1 and NCRM5 cells were cultured in STEMdiff™ APEL™ medium in the presence of GSK and TGFβ inhibitors to direct their differentiation to cells of the neural crest [28, 29]. Cells were selected for the neural crest markers CD34⁻KDR⁻CD271⁺PDGFRa⁻ by FACS and grown further on Laminin-521 and Cell Bind coated plates to induce mesenchymal cell differentiation (Fig 2). Mesenchymal cells were identified by the absence of CD45 and CD31 surface markers and the detection of CD44⁺CD73⁺CD105⁺ by flow cytometry and were transferred to osteogenic medium for differentiation [28, 40]. The pluripotent cell gene markers *OCT3/4* and *NANOG* were detected in parental iPS cells but were virtually undetectable in neural crest and mesenchymal cells verifying loss of pluripotency and evidence of cell differentiation (Fig 3A). Following differentiation, NCRM1 and NCRM5 neural crest cells expressed *GSC* and *SOX10* and mesenchymal cells expressed *FOXC2* transcripts (Fig 3A) [29, 41–43]. The *NOTCH3^(6692-93insC)* pathogenic variant did not alter the differentiation of either NCRM1 or NCRM5 cells to neural crest or mesenchymal cells (Fig 3B). Culture of mesenchymal cells under osteogenic conditions induced their differentiation toward osteoblasts and formed mineralized nodules (Figs 2 and 4) and expressed *ALPL*, *BGLAP*, *RUNX2* and low levels of *SP7*, genes associated with osteoblasts. *RUNX2* declined and *ALPL* increased as the culture progressed over a 4-week period (Fig 4). *ACP5*, an osteoclast gene marker, was detected during

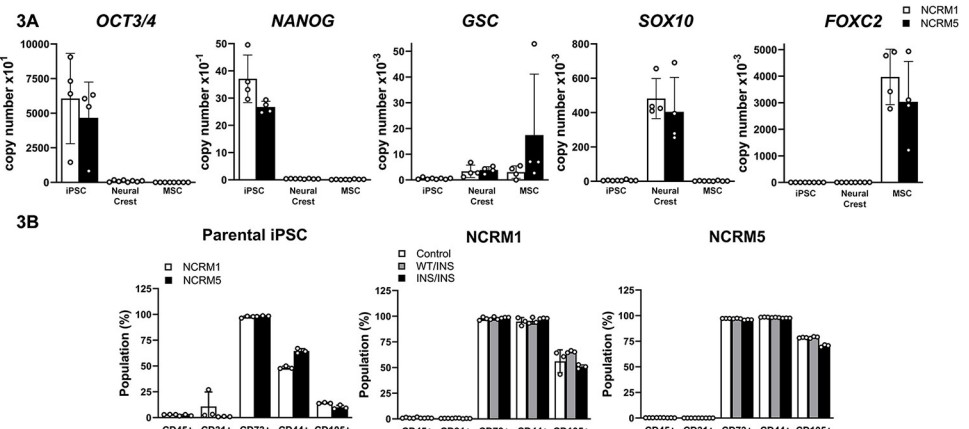

**Fig 3. NCRM1 and NCRM5 induced pluripotent stem (iPS) cells differentiate into neural crest and the mesenchymal cell lineage *in vitro*.** iPS cells were cultured it STEMdiff™ APEL™ medium in the presence of GSK and TGFβ inhibitors, selected for KDR⁻CD34⁻CD271⁺PDGFRa⁻ by FACS and cultured in the presence of ROCK and TGFβ inhibitors for 24 h. Cells were transferred to Laminin-521 and Cell Bind coated plates in the presence of NutriStem XF medium for 2 weeks. Panel A shows pluripotent cell gene markers *OCT3/4* and *NANOG*, neural crest gene markers *GSC* and *SOX10* and mesenchymal cell marker *FOXC2* mRNA expressed as copy number corrected for *HPRT* in undifferentiated parental NCRM1 (white bars) and NCRM5 (black bars) iPS cells, neural crest and mesenchymal cells (MSC). Panel B shows mesenchymal cell surface markers determined by flow cytometry of NCRM1 and NCRM5 parental cells (left) and iPS cells harboring a *NOTCH3^{6692-93insC}* insertion and isogenic controls following neural crest and mesenchymal cell differentiation. Individual values are shown, and bars and ranges represent means ± SD; n = 4 technical replicates.

the initial phase of the culture and declined during osteoblast differentiation. *NOTCH1*, *NOTCH2* and *NOTCH3*, but not *NOTCH4*, transcripts were detected in NCRM1 and NCRM5 cells during osteogenic differentiation.

## *NOTCH3^{6692-93insC}* pathogenic variant causes a modest enhancement of osteoblastogenesis *in vitro*

To determine the impact of the *NOTCH3^{6692-93insC}* insertion on osteoblast cell differentiation, homozygous NCRM1 (Fig 5) and NCRM5 (Fig 6) cells for the insertion and isogenic controls

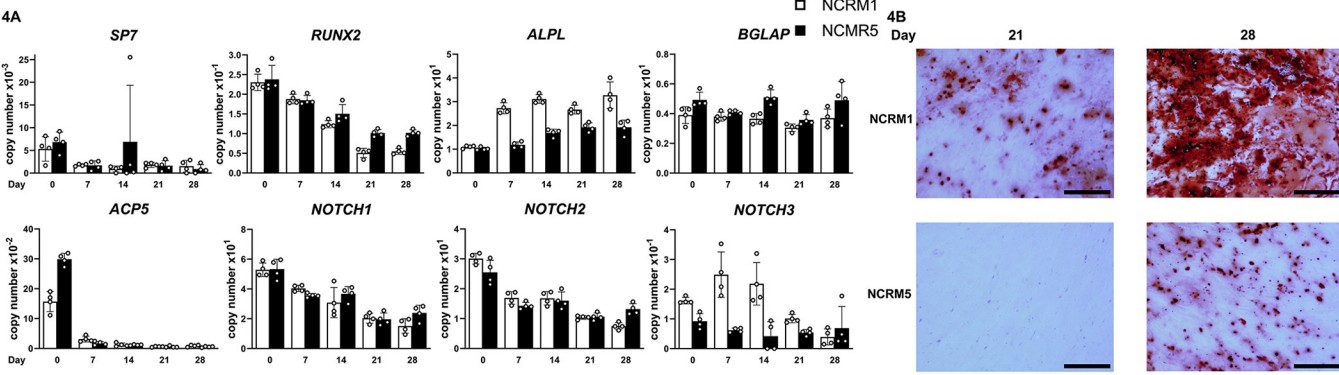

**Fig 4. NCRM1 and NCRM5 induced pluripotent stem (iPS) cells differentiate to the osteoblast lineage.** NCRM1 (white bars) and NCRM5 (black bars) iPS cells were cultured to induce neural crest and mesenchymal cell differentiation as described in Figs 2 and 3 and transferred to osteogenic medium for 4 weeks. In Panel A, *SP7*, *RUNX2*, *ALPL*, *BGLAP*, *ACP5*, *NOTCH1*, *NOTCH2* and *NOTCH3* mRNA levels were determined at the indicated times. Transcript levels are expressed as copy number following correction for *HPRT* copy number. Individual values are shown, and bars and ranges represent means ± SD; n = 4 technical replicates. In Panel B, mineralized nodules stained with alizarin red obtained after 4 weeks of culture in osteogenic medium. Bars in the right corner represent 200 μm. *NOTCH4* was not detected.

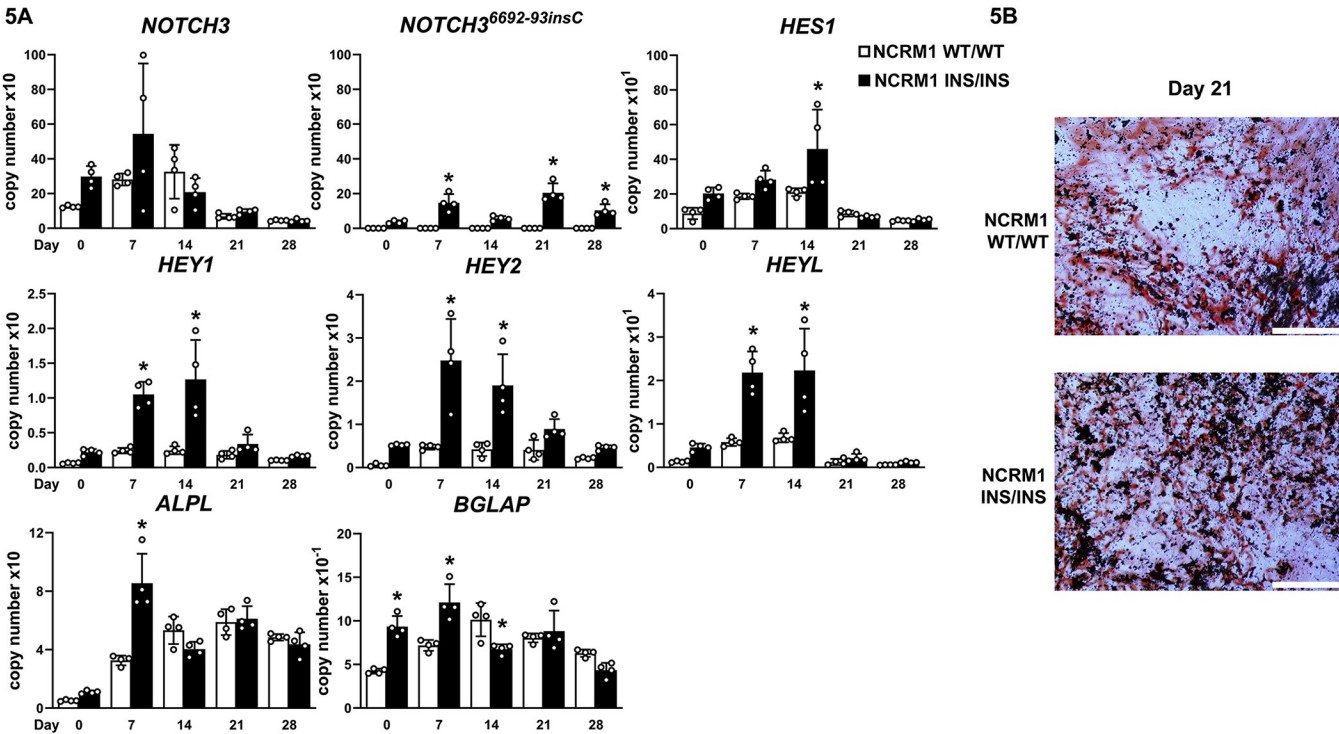

**Fig 5. $NOTCH3^{6692\_93insC}$ causes a NOTCH3 gain-of-function and a modest enhancement of osteogenesis in NCRM1 induced pluripotent stem (iPS) cells.** iPS cells harboring a $NOTCH3^{6692\_93insC}$ insertion in both alleles (INS/INS, black bars) and isogenic controls (WT/WT white bars) were cultured under conditions that induce neural crest and mesenchymal cell differentiation and transferred to osteogenic medium for 4 weeks. In Panel A, $NOTCH3$, $NOTCH3^{6692\_93insC}$, $HES1$, $HEY1$, $HEY2$, $HEYL$, $ALPL$ and $BGLAP$ mRNA levels were determined at the indicated times. Transcript levels are expressed as copy number following correction for $HPRT$ copy number. Individual values are shown, and bars and ranges represent means ± SD; n = 4 technical replicates. In Panel B, mineralized nodules stained with alizarin red after 3 weeks of culture in osteogenic medium. Bars in the right corner represent 200 μm. *Significantly different between $NOTCH3^{6692\_93insC}$ and control cells by ANOVA, $p < 0.05$.

were cultured under osteogenic conditions. $NOTCH3$ mRNA was detected in control and mutant cells throughout the culture since the RT-PCR could not discriminate a single 6692-93insC. $NOTCH3^{6692-93insC}$ transcripts were detected only in iPS cells harboring the pathogenic variant allele and as expected, the level of expression was higher in homozygous than in heterozygous $NOTCH3^{6692-93insC}$ clones (not shown). $NOTCH3^{6692-93insC}$ cells expressed higher levels of $HEY1$, $HEY2$ and $HEYL$ transcripts than isogenic controls during the first 2 weeks of culture in NCRM1 and throughout the culture period in NCRM5 mutant cells demonstrating that the $NOTCH3$ insertion resulted in a gain-of-Notch function (Figs 5 and 6). Accordingly, there was a short-lived induction of $ALPL$ and $BGLAP$ in NCRM1 mutant cells (Fig 5) and a more sustained increased expression of $ALPL$ and $BSP$ in NCRM5 cells harboring the $NOTCH3^{6692-93insC}$ insertion compared to isogenic controls (Fig 6). There was greater formation of mineralized nodules in $NOTCH3^{6692-93insC}$ NCRM1 and NCRM5 cells than in isogenic control cells (Figs 5 and 6). $WNT1$ was not detected in either wild type or $NOTCH3^{6692\_93insC}$, NCRM1 or NCRM5 iPS cells.

## NOTCH3 ASOs downregulate NOTCH3 expression in mesenchymal and osteogenic cells

To determine the effect of NOTCH3 ASOs in control and $NOTCH3^{6692-93insC}$ cells, ASOs targeting $NOTCH3$ and the $NOTCH3^{6692-93insC}$ pre-mRNA were designed and synthesized (Fig 7).

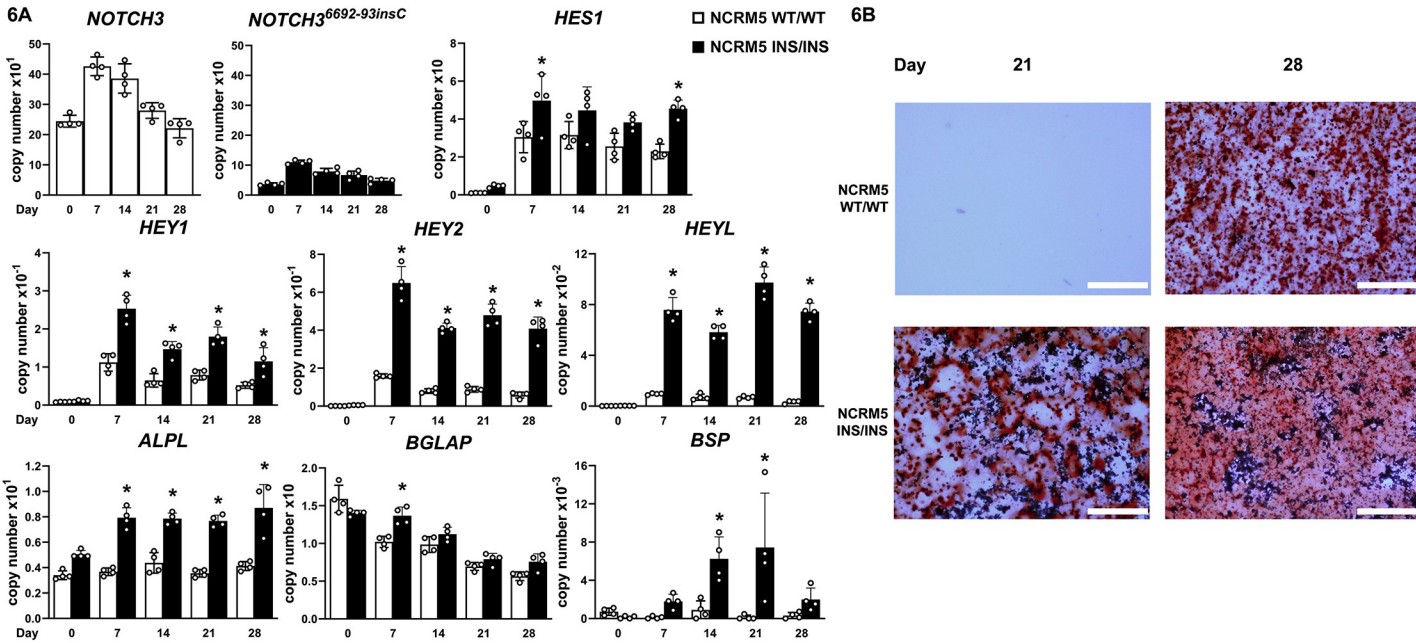

**Fig 6. NOTCH3⁶⁶⁹²_⁹³ⁱⁿˢᶜ causes a NOTCH3 gain-of-function and a modest enhancement of osteogenesis in NCRM5 induced pluripotent stem (iPS) cells.** iPS cells harboring a *NOTCH3*⁶⁶⁹²_⁹³ⁱⁿˢᶜ insertion in both alleles (INS/INS, black bars) and isogenic controls (WT/WT, white bars) were cultured under conditions that induce neural crest and mesenchymal cell differentiation and transferred to osteogenic medium for 4 weeks. In Panel A, *NOTCH3* (in isogenic controls), *NOTCH3*⁶⁶⁹²_⁹³ⁱⁿˢᶜ (in *NOTCH3* mutant iPS cells), *HES1*, *HEY1*, *HEY2*, *HEYL*, *ALPL*, *BGLAP* and *BSP* mRNA levels were determined at the indicated times. Transcript levels are expressed as copy number following correction for *HPRT* copy number. Individual values are shown, and bars and ranges represent means ± SD; n = 4 technical replicates. In Panel B, mineralized nodules stained with alizarin red obtained following 3 and 4 weeks of culture in osteogenic medium. Bars in the right corner represent 200 μm. *Significantly different between *NOTCH3*⁶⁶⁹²_⁹³ⁱⁿˢᶜ and control cells by ANOVA, $p < 0.05$.

The effect of NOTCH3 ASOs was tested initially in undifferentiated iPS *NOTCH3*⁶⁶⁹²⁻⁹³ⁱⁿˢᶜ and isogenic control cells, but neither the addition of NOTCH3 nor NOTCH3 mutant ASOs to the culture medium downregulated *NOTCH3* or *NOTCH3*⁶⁶⁹²⁻⁹³ⁱⁿˢᶜ transcripts, possibly because of limited ASO internalization by undifferentiated iPS cells (not shown). In subsequent

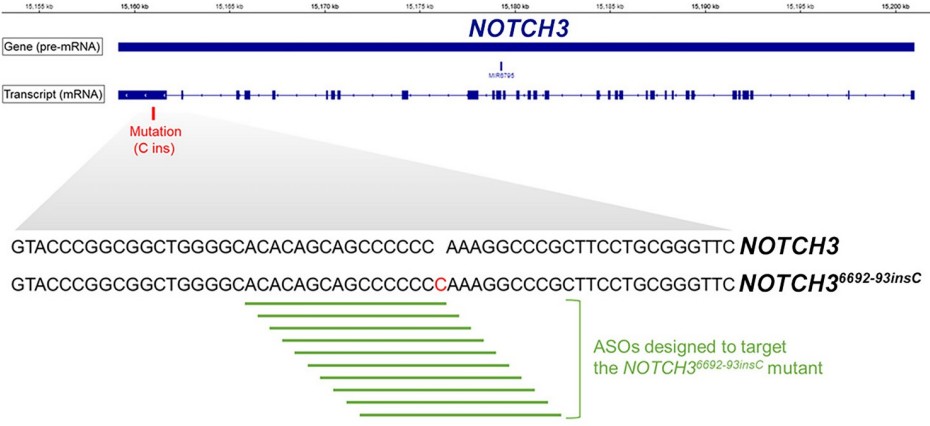

**Fig 7. Ten antisense oligonucleotides (ASOs) targeting the NOTCH3⁶⁶⁹²_⁹³ⁱⁿˢᶜ pre-mRNA were designed to degrade the mutant transcript.** Green lines represent the site targeted by each ASO. The colored letter represents the *NOTCH3*⁶⁶⁹²_⁹³ⁱⁿˢᶜ insertion harbored by mutant cells compared to wild type human *NOTCH3*. The wild type human *NOTCH3* pre-mRNA sequence is depicted at the top of the figure.

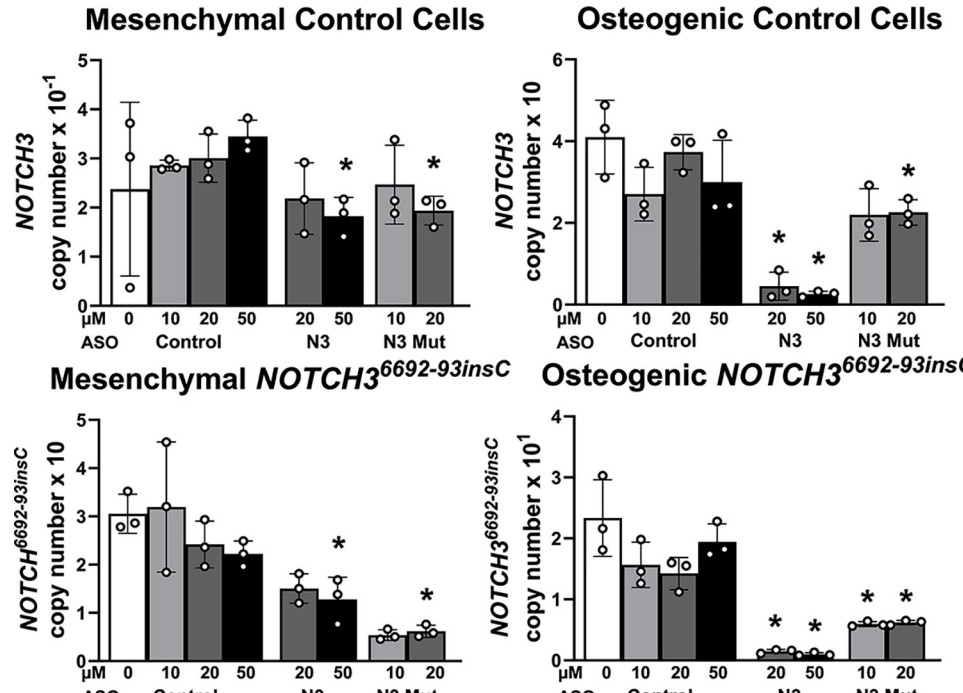

**Fig 8. NOTCH3 and NOTCH3 mutant antisense oligonucleotides (ASOs) downregulate *NOTCH3*$^{6692-93insC}$ mutant transcripts.** Control and NOTCH3 ASOs targeting *NOTCH3* or the *NOTCH3*$^{6692\_93insC}$ insertion were tested for their effects on NOTCH3 and *NOTCH3*$^{6692\_93insC}$ mRNA expression in NCRM1 mesenchymal cells before (left) and following the exposure to osteogenic medium for 14 days (right). *NOTCH3* and *NOTCH3*$^{6692\_93insC}$ mRNA levels were obtained 72 h after the addition of control, NOTCH3 or NOTCH3 mutant ASOs at the indicated concentrations to the culture medium. Transcript levels are expressed as *NOTCH3* (upper panels) and *NOTCH3*$^{6692-93insC}$ (lower panels) copy number following correction for *HPRT* copy number. Individual values are shown, and bars and ranges represent means ± SD n = 3 technical replicates. *Significantly different between NOTCH3 ASO or mutant NOTCH3 ASO and control ASO by unpaired *t* test, *p* < 0.05.

experiments, NOTCH3 and NOTCH3 mutant ASOs were tested in mesenchymal and following culture under osteogenic conditions for 2 weeks (osteogenic cells). NOTCH3 ASOs added to the culture medium at 20 and 50 μM for 72 h downregulated *NOTCH3* and *NOTCH3*$^{6692-93insC}$ transcripts to an equal extent (30–40%) in mesenchymal and ~90% in osteogenic cells (Fig 8). The addition of one of the 10 ASOs designed to target mutant *NOTCH3* to the culture medium of mesenchymal cells downregulated *NOTCH3*$^{6692-93insC}$ mRNA and downregulated *NOTCH3* wild type mRNA modestly indicating specificity for the *NOTCH3* insertion (Fig 8). The NOTCH3 mutant ASO at 10 μM and 20 μM downregulated *NOTCH3*$^{6692-93insC}$ by 70–80% in mesenchymal cells and by 44–55% in osteogenic cells. The effect was more pronounced for *NOTCH3*$^{6692-93insC}$ than for NOTCH3 wild type mRNA, which was downregulated by 15–30% in mesenchymal cells and 20–40% in osteogenic cells. As expected, the *NOTCH3*$^{6692-93insC}$ transcript was not detected in control wild type cells. To verify the activity of NOTCH3 ASOs, they were tested for their effects on canonical targets of Notch signaling in osteogenic cells. *HEY1*, *HEY2*, and *HEYL* mRNA expression was increased in *NOTCH3*$^{6692-93insC}$ cells, and the effect was reversed by the ASO targeting NOTCH3 at 50 μM and the ASO targeting *NOTCH3*$^{6692-93insC}$ at 20 μM (Fig 9). However, some of these changes did not reach statistical significance due to a small sample size (n = 3) and variability. The results suggest a reversal of the enhanced signal activation observed in LMS mutant cells.

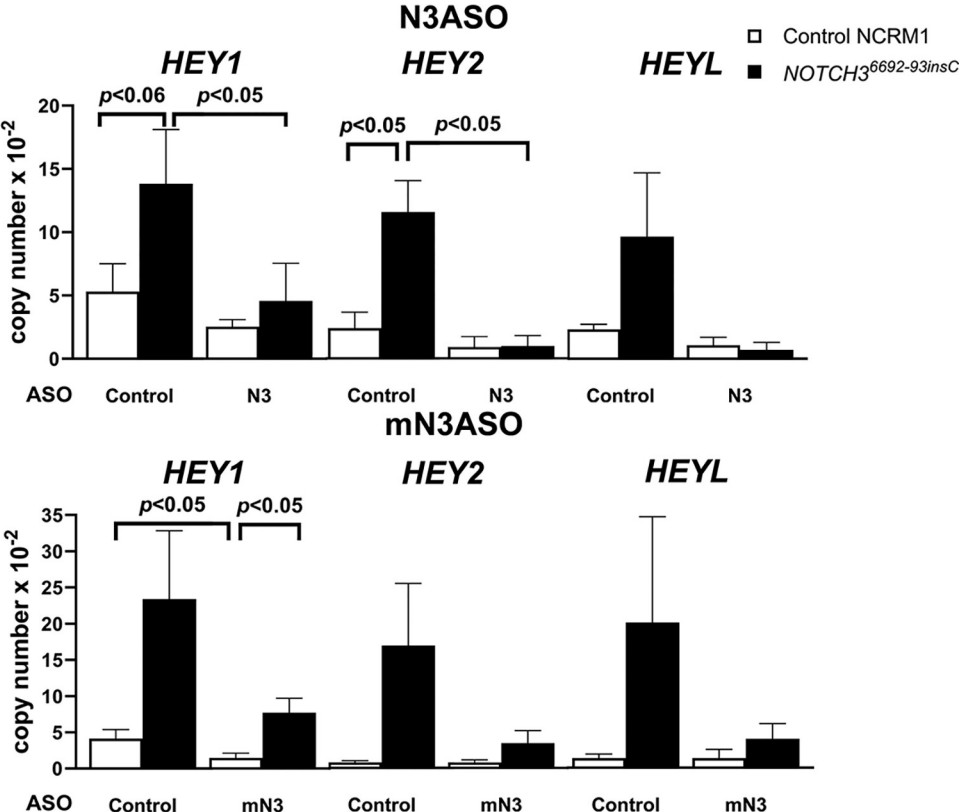

**Fig 9. NOTCH3 and NOTCH3 mutant antisense oligonucleotides (ASOs) downregulate Notch target genes.**
Control and NOTCH3 ASOs targeting *NOTCH3* (N3) at 50 µM or the *NOTCH3$^{6692\_93insC}$* insertion (mN3) at 20 µM were tested for their effects on *HEY1*, *HEY2* and *HEYL* mRNA expression in NCRM1 mesenchymal cells exposed to osteogenic conditions for 2 weeks 72 h after ASO addition to the culture. Control ASO was added at the same concentration. Transcript levels are expressed as *HEY1*, *HEY2* and *HEYL* copy number following correction for *HPRT* copy number following treatment with NOTCH3 (upper panel) or NOTCH3 mutant ASO (lower panel). Individual values are shown, and bars and ranges represent means ± SD; n = 3 technical replicates. Bars and *p* values indicate significant differences between *NOTCH3$^{6692\_93insC}$* and control cells, and between NOTCH3 ASO or mutant NOTCH3 ASO and control ASO by ANOVA.

## Discussion

The present studies were conducted to determine whether a *NOTCH3* pathogenic variant found in LMS or Lehman Syndrome had a phenotypic impact on mesenchymal and osteogenic cell differentiation. In addition, the work tested whether *NOTCH3* and its pathogenic variant could be downregulated in human cells *in vitro* by the administration of ASOs. The current findings confirm in human iPS cells that a pathogenic variant harbored in LMS results in a gain-of-Notch function as evidenced by increased expression of the canonical target genes *HES1*, *HEY1*, *HEY2* and *HEYL*. The results confirm observations reported in mouse models of LMS termed *Notch3$^{em1Ecan}$* that presented with a NOTCH3 gain-of-function and osteopenia secondary to an increase in receptor activator of NF-κB ligand (RANKL) in cells of the osteo-blast lineage [38]. The number and function of osteoblasts was not affected in *Notch3$^{em1Ecan}$* mice and *in vitro* experiments did not reveal either enhanced or suppressed osteoblastogenesis. In contrast to these findings, iPS cells harboring an LMS pathogenic variant exhibited a modest enhancement in osteogenesis as evidenced by an increase in mineralized nodule formation and select gene markers associated with the osteoblast phenotype. The effect was more evident and sustained in NCRM5 than in NCRM1 cells harboring the *NOTCH3* pathogenic variant,

but the reasons for the difference were not explored. NCRM1 cells harbor a 602.9 kilobase gain in chromosome 20 Band 20q11.21-20q11.21 position 29, 804, 293–30, 407, 232 although whether this could influence their phenotype is unknown. The modest discrepancy between human and mouse cells might be related to different culture models since the cells from *Notch3$^{em1Ecan}$* mice were studied in primary cultures following the isolation of relatively mature cells of the osteoblast lineage [38]. It is possible that the outcome observed in iPS cells is due to an effect of NOTCH3 in an immature, less differentiated cell or to the fact that often the effects of Notch are cell-context dependent.

The current work also was undertaken to determine whether ASOs could be developed to target and downregulate a *NOTCH3* pathogenic variant in iPS cells. One of the ASOs tested downregulated the *NOTCH3$^{6692-93insC}$* with a more modest effect on wild type *NOTCH3* transcripts in mesenchymal and osteogenic cell cultures indicating a degree of specificity. In addition, ASOs targeting either wild type or mutant *NOTCH3* alleles decreased the induction of canonical Notch target genes observed in *NOTCH3$^{6692-93insC}$* cells. Although some of the changes did not achieve statistical significance due to a small sample size, the results would suggest at least a partial reversal of the NOTCH3 gain-of-function observed in cells harboring the LMS pathogenic variant and provide evidence of a biological effect. It was not possible to test the long-term effects of NOTCH3 ASOs on the osteogenic phenotype of *NOTCH3$^{6692-93insC}$* cells because the phenotype was modest and the prolonged exposure of cells to the ASO resulted in cellular toxicity.

Various approaches have been reported to downregulate signaling by Notch receptors. However, often they lack specificity to Notch activity or to a specific Notch receptor. Antibodies targeting the negative regulatory region (NRR) of Notch prove to be an exception and in previous work, we demonstrated that anti-NOTCH2 NRR and anti-NOTCH3 NRR antibodies reverse skeletal phenotypic manifestations of *Notch2$^{tm1.1Ecan}$*, a mouse model of Hajdu Cheney Syndrome (HCS) and of *Notch3$^{em1Ecan}$*, a mouse model of LMS [44, 45]. A limitation of anti-Notch NRR antibodies is the fact that they do not discriminate between the wild type and mutant forms of the receptor and by causing a substantial downregulation of Notch signaling their administration may result in gastrointestinal toxicity.

Previously, we have shown the *in vitro* and *in vivo* ability of Notch2 ASOs to downregulate *Notch2* expression and as a result improve the osteopenia of mice harboring a *Notch2* gain-of-function mutation found in HCS [46]. We also reported that ASOs targeting either *Notch3* or *Notch3* mutants replicating the pathogenic variant of LMS, inhibit *Notch3* mRNA and improve the cortical osteopenia found in *Notch3$^{em1Ecan}$* mice [23, 24]. The present work confirms the feasibility of the approach and extends the findings to a human, albeit *in vitro*, model.

There are limitations in the present work. Phenotypic alterations of experimental and control iPS cells and testing of ASOs were conducted *in vitro* for markers of osteogenesis and no assays were conducted to verify bone formation *in vivo* [47]. The enhancement of osteoblastogenesis by the *NOTCH3* LMS pathogenic variant was modest. The phenotypic impact of the *NOTCH3$^{6692-93insC}$* variant and the effects of Notch3 ASOs were assessed in cells of the osteoblast lineage and not in the myeloid/osteoclast lineage because *NOTCH3* is not expressed in this lineage either in human or murine cells and NOTCH3 does not have direct effects on osteoclastogenesis [6, 39]. Indeed, *NOTCH1* and *NOTCH2* transcripts are detected in iPS cells differentiating toward the myeloid/osteoclast lineage whereas low/undetectable levels of *NOTCH3* and *NOTCH4* are found (Canalis, et al., unpublished observations). Because only iPS cells derived from umbilical blood of male subjects were studied, one should not extrapolate the results observed to iPS cells derived from female individuals or iPS cells from other sources.

In conclusion, a *NOTCH3* pathogenic variant found in LMS causes a modest enhancement of osteogenesis in iPS cells *in vitro*, and ASOs can be used to target and downregulate *NOTCH3* and an LMS pathogenic variant in human iPS cells.

## Acknowledgments

The authors thank Drs. Pamela Robey and Fahad Kidwai for helpful advice, Magda Mocarska and Emily Denker for technical assistance, Mary Yurczak for secretarial support, and Emily Denker for creating figures for the manuscript.

## Author Contributions

**Conceptualization:** Ernesto Canalis.

**Funding acquisition:** Ernesto Canalis.

**Investigation:** Jungeun Yu, Lauren Schilling.

**Methodology:** Ernesto Canalis.

**Project administration:** Ernesto Canalis.

**Resources:** Paymaan Jafar-nejad, Michele Carrer.

**Supervision:** Ernesto Canalis.

**Validation:** Ernesto Canalis.

**Visualization:** Lauren Schilling, Paymaan Jafar-nejad, Michele Carrer.

**Writing – original draft:** Ernesto Canalis.

**Writing – review & editing:** Ernesto Canalis, Jungeun Yu, Paymaan Jafar-nejad, Michele Carrer.

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
