## [Decision Letter · Decision Letter 0]

21 Aug 2024

PONE-D-24-20556A NOTCH3 Pathogenic Variant Influences Osteogenesis and can be Targeted by Antisense Oligonucleotides in Induced Pluripotent Stem CellsPLOS ONE

Dear Dr. Canalis,

Thank you for submitting your manuscript to PLOS ONE. After careful consideration, we feel that it has merit but does not fully meet PLOS ONE’s publication criteria as it currently stands. Therefore, we invite you to submit a revised version of the manuscript that addresses the points raised during the review process

We look forward to receiving your revised manuscript.

Kind regards,

Md Shaifur Rahman, Ph.D

Academic Editor

PLOS ONE

Journal Requirements:

Antisense oligonucleotides targeting a NOTCH3 mutation in male mice ameliorate the cortical osteopenia of lateral meningocele syndrome - https://doi.org/10.1016/j.bone.2023.116898

(among others)

In your revision ensure you cite all your sources (including your own works), and quote or rephrase any duplicated text outside the methods section. Further consideration is dependent on these concerns being addressed.

   "This work was supported by the National Institute of Arthritis and Musculoskeletal and Skin Diseases (NIAMS) [AR076747 (EC) and AR072987 (EC)].  The content is solely the responsibility of the authors and does not necessarily represent the official views of the National Institutes of Health."

   "EC, JY, LS have no competing interests; PJ and MC are paid employees of Ionis Pharmaceuticals." 

We note that one or more of the authors are employed by a commercial company: Ionis Pharmaceuticals 

6. We note that your Data Availability Statement is currently as follows: All relevant data are within the manuscript and its Supporting Information files.

Reviewers' comments:

Reviewer's Responses to Questions

**Comments to the Author**

1. Is the manuscript technically sound, and do the data support the conclusions?

Reviewer #1: Yes

2. Has the statistical analysis been performed appropriately and rigorously? 

Reviewer #1: Yes

3. Have the authors made all data underlying the findings in their manuscript fully available?

Reviewer #1: Yes

4. Is the manuscript presented in an intelligible fashion and written in standard English?

Reviewer #1: Yes

5. Review Comments to the Author

Reviewer #1: The authors present an interesting approach to simulate LMS in iPS cells. Their differentiation towards an osteogenic linage is analyzed. Antisense oligonucleotids were used to counter the NOTCH3 gain-of-funtion mutation in LMS iPS cells.

The work is a novel approach and of particular interest, since the same group already analyzed LMS in a mouse model in depth. Thus, I think it is a great step by the group to further investigate LMS in human cells, now. iPS cells have not been frequently used by bone researches and bear great potential. The protocols and experiments are in detail explained and are understandable to the reader. I have only very minor comments, which the authors might re-think:

- introduction: mention the incidence of LMS

- Table 1: Did you also analyze WNT1? - it interaction with the Notch pathway is well described. It would be interesting to see at least how it is indirectly influenced by a NOTCH3 overexpression.

- only osteoblastic genes were measured in qRT-PCR. At least one osteoclastic marker gene (ACP5, OPG or CD68...) would be interesting

- The figures are in very bad quality. I guess this is due to the editorialmanager program, otherwise resolution must be improved

- Figure 2: add a scale bar to the microscopic pictures, it is not clear at which the resolution the images are taken

- Figure 5: I recommend to add a title or give at least the hint in the figure itself that it is NCRM1 cells and Figure 6 is NCRM5. Otherwise, the almost alike looking figures are confusing to the reader.

-Figure 5/6: Level the axis/Y-bar of Notch3 Wildtype and Notch3 Mutant. The first bar only goes up to 10x10^1 and the mutant´s bar to 30. By leveling the bar, gain-of-function mechanism is straight to see at the first glance

Figure 9: The usage of * and # is awkward. I recommend bars, connecting the columns which are compared. This is by far clearer to the reader.

6. PLOS authors have the option to publish the peer review history of their article (what does this mean?). If published, this will include your full peer review and any attached files.

Reviewer #1: No

---

## [Author Response · Author response to Decision Letter 0]

29 Aug 2024

August 29, 2024

Md Shaifur Rahman, Ph.D.

Academic Editor

PLOS ONE

Dear Dr. Rahman:

In accordance with your letter of August 21, 2024, we would like to submit a revised version of the manuscript entitled “A NOTCH3 Pathogenic Variant Influences Osteogenesis and can be Targeted by Antisense Oligonucleotides in Induced Pluripotent Stem Cells” (PONE-D-24-20556) to be considered for publication by PLOS ONE. 

Journal Requirements: 

 PLOS ONE style requirements have been ensured.

2. We noticed you have some minor occurrence of overlapping text with the following previous publication(s), which needs to be addressed - https://doi.org/10.1016/j.bone.2023.116898.

 We have compared the current work with our previous publication and whenever possible have corrected the minor occurrences of overlapping.

3. Please note that funding information should not appear in any section or other areas of your manuscript. We will only publish funding information present in the Funding Statement section of the online submission form.

 Please remove any funding-related text from the manuscript.

 Funding information was removed from the text of the manuscript.

4. Thank you for stating the following financial disclosure: "This work was supported by the National Institute of Arthritis and Musculoskeletal and Skin Diseases (NIAMS) [AR076747 (EC) and AR072987 (EC)]. The content is solely the responsibility of the authors and does not necessarily represent the official views of the National Institutes of Health."

 On our behalf, we request to update the Role of Funder Statement to: 

 “This work was supported by the National Institute of Arthritis and Musculoskeletal and Skin Diseases (NIAMS) [AR076747 (EC) and AR072987 (EC)]. The content is solely the responsibility of the authors and does not necessarily represent the official views of the National Institutes of Health. The funding institute had no role in study design, data collection and analysis, decision to publish, or preparation of the manuscript.

 Ionis Pharmaceuticals [PJ and MC] provided antisense oligonucleotides and guidance for the use of the antisense oligonucleotides, PJ and MC discussed the study design with other authors but did not have any additional role in data collection and analysis for, or initial preparation of, the manuscript. The specific roles of these authors are articulated in the ‘author contributions’ section.”

5. Thank you for stating the following in the Competing Interests section: "EC, JY, LS have no competing interests; PJ and MC are paid employees of Ionis Pharmaceuticals."

We note that one or more of the authors are employed by a commercial company: Ionis Pharmaceuticals

a. Please provide an amended Funding Statement declaring this commercial affiliation, as well as a statement regarding the Role of Funders in your study.

 Under Funding Statement, we now include “Ionis Pharmaceuticals [PJ and MC] provided antisense oligonucleotides and guidance for the use of the antisense oligonucleotides, and discussed the study design with other authors but did not have any additional role in data collection and analysis for, or the initial preparation of, the manuscript. The specific roles of these authors are articulated in the ‘author contributions’ section.”

The roles of all authors have been verified.

 Updated Funding Statement is included under 4. 

 Please update Competing Interest Statement as follows:

PJ and MC are paid employees of Ionis Pharmaceuticals. Please note that the synthesis and applications of antisense oligonucleotides may be covered by patent(s) filed by Ionis Pharmaceuticals. Individuals wanting to obtain antisense oligonucleotides from Ionis Pharmaceuticals are required to contact Ionis Pharmaceuticals directly. 

This does not alter our adherence to PLOS ONE policies on sharing data and materials.” as detailed in http://journals.plos.org/plosone/s/competing-interests

6. We note that your Data Availability Statement is currently as follows: All relevant data are within the manuscript and its Supporting Information files.

 All figures contain individual actual values used to calculate means ± SD. Images are representative images of the culture, and no specific points were extracted for their creation.

 Please note that the synthesis and applications of antisense oligonucleotides may be covered by patent(s) filed by Ionis Pharmaceuticals.

7. Please review your reference list to ensure that it is complete and correct. 

 Reference list has been reviewed.

Reviewer #1:

The authors present an interesting approach to simulate LMS in iPS cells. Their differentiation towards an osteogenic linage is analyzed. Antisense oligonucleotides were used to counter the NOTCH3 gain-of-function mutation in LMS iPS cells.

The work is a novel approach and of particular interest, since the same group already analyzed LMS in a mouse model in depth. Thus, I think it is a great step by the group to further investigate LMS in human cells, now. iPS cells have not been frequently used by bone researchers and bear great potential. The protocols and experiments are in detail explained and are understandable to the reader. I have only very minor comments, which the authors might re-think:

- introduction: mention the incidence of LMS

 The incidence of LMS is unknown and less than 100 cases have been reported. This is now stated in the introduction, end of paragraph 1, line 49.

- Table 1: Did you also analyze WNT1? - it interaction with the Notch pathway is well described. It would be interesting to see at least how it is indirectly influenced by a NOTCH3 overexpression.

 As requested, we performed RT-PCR for WNT1 in control and NOTCH3 mutant cells. Please see Table 1. WNT1 was not detected in either wild type or NOTCH3 mutant cells and this is stated in the text of the revised manuscript (page 15, paragraph 1, line 273).

- only osteoblastic genes were measured in qRT-PCR. At least one osteoclastic marker gene (ACP5, OPG or CD68...) would be interesting

 We now include data on the expression of the osteoclast gene marker ACP5 in Figure 4 and cite the information in the text in page 13, paragraph 1, line 231.

- The figures are in very bad quality. I guess this is due to the editorial manager program, otherwise resolution must be improved

 Unfortunately, the quality of the figures declines when the paper is uploaded into the PLOS ONE website and the solution rests with the publisher.

- Figure 2: add a scale bar to the microscopic pictures, it is not clear at which the resolution the images are taken.

 A scale bar is now added to the pictures in Figure 2, and images in all figures of the paper.

- Figure 5: I recommend to add a title or give at least the hint in the figure itself that it is NCRM1 cells and Figure 6 is NCRM5. Otherwise, the almost alike looking figures are confusing to the reader.

 We now identify NCRM1 or NCRM5 cells in the body of each figure.

-Figure 5/6: Level the axis/Y-bar of Notch3 Wildtype and Notch3 Mutant. The first bar only goes up to 10x10^1 and the mutant´s bar to 30. By leveling the bar, gain-of-function mechanism is straight to see at the first glance.

 We now show copy number x 1010 for both NOTCH3 and NOTCH36692-93insC in Figure 5. In Figure 6, we show copy number x 101 for NOTCH3 and copy number x10 for NOTCH36692-93insC otherwise bars could not be seen for the mutant transcript at the same y-axis scale.

Figure 9: The usage of * and # is awkward. I recommend bars, connecting the columns which are compared. This is by far clearer to the reader.

 * and # were removed and we now include lines connecting the columns being compared.

Sincerely yours, 

Ernesto Canalis, M.D.

Professor of Orthopaedics and Medicine

Director, Center for Skeletal Research

Co-Director, Musculoskeletal Institute

---

## [Decision Letter · Decision Letter 1]

16 Dec 2024

A NOTCH3 Pathogenic Variant Influences Osteogenesis and can be Targeted by Antisense Oligonucleotides in Induced Pluripotent Stem Cells

PONE-D-24-20556R1

Dear Prof. Dr. Canalis,

We’re pleased to inform you that your manuscript has been judged scientifically suitable for publication and will be formally accepted for publication once it meets all outstanding technical requirements.

Kind regards,

Md Shaifur Rahman, Ph.D

Academic Editor

PLOS ONE

---

## [Editor Report · Acceptance letter]

20 Dec 2024

PONE-D-24-20556R1 

PLOS ONE

Dear Dr. Canalis, 

I'm pleased to inform you that your manuscript has been deemed suitable for publication in PLOS ONE. Congratulations! Your manuscript is now being handed over to our production team.

Kind regards, 

on behalf of

Dr. Md Shaifur Rahman 

Academic Editor

PLOS ONE